# *Crassoascoma* gen. nov. (*Lentitheciaceae*, *Pleosporales*): Unrevealing Microfungi from the Qinghai-Tibet Plateau in China

**Zuo-Peng Liu** [1,†], **Sheng-Nan Zhang** [1,2,†], **Ratchadawan Cheewangkoon** [2], **Qi Zhao** [3] and **Jian-Kui Liu** [1,2,*]

[1] School of Life Science and Technology, Center for Informational Biology, University of Electronic Science and Technology of China, Chengdu 611731, China; perr_bio@163.com (Z.-P.L.); zshengnanbio@gmail.com (S.-N.Z.)

[2] Department of Entomology and Plant Pathology, Faculty of Agriculture, Chiang Mai University, Chiang Mai 50200, Thailand; ratchadawan.c@cmu.ac.th

[3] Key Laboratory for Plant Diversity and Biogeography of East Asia, Kunming Institute of Botany, Chinese Academy of Sciences, Kunming 650201, China; zhaoqi@mail.kib.ac.cn

\* Correspondence: Liujiankui@uestc.edu.cn; Tel.: +86-028-6183-1832

† These authors have contributed equally to this work and share first authorship.

**Abstract:** Microfungi associated with woody plants on the Qinghai–Tibet Plateau (QTP) were investigated, and four collections associated with *Potentilla fruticosa* were obtained from Gansu and Qinghai provinces in China. Morphologically, they line well with *Lentitheciaceae* in having subglobose to globose ascomata with brown setae on the papilla but can be distinguished from other genera by its superficial, globose, black, thick-walled ascomata, and fusiform, hyaline (rarely pale brown), one-septate ascospores, surrounded by an entire mucilaginous sheath. The phylogenetic analyses based on a combined SSU, ITS, LSU and *TEF*1-α sequence data showed that four isolates formed a monophyletic clade among the genera of *Lentitheciaceae*, and present as a distinct lineage (sister clade to *Darksidea*). Therefore, we introduce a new genus *Crassoascoma*, with *C. potentillae* as the type to accommodate these taxa. Detailed description and illustration are provided, and the establishment of new taxa is justified with morphology and phylogenetic evidence.

**Keywords:** two new taxa; *Dothideomycetes*; high-elevation; phylogeny; taxonomy

## 1. Introduction

*Dothideomycetes* is the largest class of the phylum Ascomycota, comprising 38 orders and 210 families (including the *incertae sedis*) [1,2]. The morphology of this class is diverse and usually have bitunicate asci in their sexual morphs. *Pleosporales* is the largest order in *Dothideomycetes*, which includes approximately 91 families and 606 genera (including 48 genera *incertae sedis*) [3]. However, the numbers are constantly increasing with the establishment of new taxa [4–7].

The pleosporalean family *Lentitheciaceae* was introduced by Zhang et al. [8] based on multi-gene phylogeny, of which the well-supported *Lentitheciaceae* clade accommodated the generic type *Lentithecium fluviatile*, as well as *L. arundinaceum*, *Keissleriella cladophila*, *Wettsteinina lacustris* (CBS 618.86, morphologically could not be well-verified), two bambusicolous species *Katumotoa bambusicola*, *Ophiosphaerella sasicola*, and a possibly asexual morph species *Stagonospora macropycnidia* (OSC 100965/CBS 114202). With the continuous addition of new members, the family currently comprises fourteen genera: *Darksidea* [9], *Halobyssothecium* [10], *Katumotoa* [11], *Keissleriella* [12], *Lentithecium* [8], *Murilentithecium* [13], *Neoophiosphaerella* [14], *Phragmocamarosporium* [15], *Pleurophoma* [16], *Poaceascoma* [17], *Pseudomurilentithecium* [18], *Setoseptoria* [19], *Tingoldiago* [20] and *Towyspora* [21]. Members of *Lentitheciaceae* are characterized by lenticular to globose ascomata with brown setae or glabrous, cylindrical to clavate asci with short pedicels, and morphologically diverse ascospores, mostly narrow to broad fusiform, hyaline to brown, 1–3-septate (aseptate or



muriform in some species), sometimes filiform, fasciculate, surrounded by an entire mucilaginous sheath or fusiform gelatinous sheath; the asexual morphs are stagonospora-like or dendrophoma-like [1]. Most genera have sexual morphs except for *Phragmocamarosporium* [15] and *Towyspora* [21], which only comprise coelomycetous asexual morphs. Species in *Lentitheciaceae* are generally saprobic on stems and twigs of herbaceous and woody plants in freshwater, terrestrial, marine environments [13,15,22–25] and endophytic *Darksidea* species have been isolated from semiarid habitat [9].

The Qinghai–Tibet Plateau (QTP) is the world's largest and highest plateau with an average elevation above 4000 m. Because of its complex topography and ecological environment, the QTP is considered to be one of the biodiversity hotspots with rich biological resources. In a review of the investigation of fungi in Tibet [26], a total of 2559 species distributed in 185 families and 551 genera were counted based on published records. The Basidiomycota accounted for 66.5% of the total, followed by Ascomycota, which accounted for 29.5%, while Glomeromycota, Zygomycota and Chytridiomycota were relatively few, accounted for 2.7%, 0.8%, and 0.5%, respectively [26]. This result is probably due to previous scientific expeditions that were mainly focused on macrofungi. However, it also indicated that the research on the identification of fungi on the QTP, especially the microfungi, is limited.

This study investigated microfungi associated with decayed woody plants on the QTP. Four collections with superficial, globose, black, thick-walled ascomata and fusiform, hyaline (rarely pale brown), 1-septate ascospores surrounded by mucilaginous sheaths, were identified based on morphology and multi-gene phylogeny. A novel ascomycete *Crassoascoma potentillae* was recognized and illustrated, and a new genus *Crassoascoma* (*Lentitheciaceae*, *Pleosporales*) was introduced to accommodate the new taxon *C. potentillae*.

## 2. Materials and Methods

### 2.1. Collection, Isolation and Morphological Examination

Decayed branches of *Potentilla fruticosa* L. (*Rosaceae*) were collected at an altitude over 2000 m on the QTP in 2020. The samples were placed in paper envelopes and taken to the laboratory. Fungal ascomata on the host surface were examined by using a Nikon eclipse NI-ss stereoscope. Freehand sections of ascomata were made into slides mounted in water. Photomicrographs were taken by Nikon SMZ800N stereo microscope fitted with Nikon ECLIPSE Ni upright microscope with DIC objectives connected to Nikon DS-Ri2 digital camera. The measurements of ascomata, asci, and ascospores were taken by NIS-Elements Analysis D 5.21. The photo plate was processed by Adobe Photoshop CS6 software (Adobe System Inc., San Jose, CA, USA).

Single spore isolation used the same spotted pour technique as Senanayake et al. [27], and the germinated ascospores were transferred into potato dextrose agar (PDA). The colony morphology was examined after four weeks of incubation at 25 °C, and the Methuen handbook of colour was used for colour description [28].

Herbarium specimens were deposited in the herbarium of Cryptogams Kunming Institute of Botany Academia Sinica (KUN-HKAS), Kunming, China, and Herbarium, University of Electronic Science and Technology (HUEST). The strains isolated in this study were deposited in China General Microbiological Culture Collection Center (CGMCC) and the University of Electronic Science and Technology Culture Collection (UESTCC). Taxonomic novelties were registered in MycoBank [29].

### 2.2. DNA Extration, PCR Amplification and Sequencing

Fungal mycelia of 28d colonies were scraped by sterile lancet and transferred into a 1.5 mL Eppendorf tube. DNA was extracted by using Ezup Column Fungi Genomic DNA Purification Kit (Sangon Biotech (Shanghai), Co., Ltd., Shanghai, China). The small subunit rDNA (SSU), internal transcribed spaces (ITS), large subunit rDNA (LSU), and translated elongated factor 1-alpha (*TEF*-1α) gene were amplified with universal primers NS4/NS1 [30], ITS5/ITS4, LR0R/LR5 [31], EF1-983F/EF1-2218R [32], respectively. All

polymerase chain reactions (PCR) were reacted in the volume of 25 μL mixture containing 12.5 μL 2 × PCR Master Mix (Sangon Biotech (Shanghai), Co., Ltd., China), 8.5 μL ddH$_2$O, 1 μL of each primer and 2 μL DNA template. PCR thermal cycles for four genes were performed as the following reaction conditions: initial 95 °C for 3 min, followed by 35 cycles of denaturation at 95 °C for 30 sec, elongation at 72 °C for 30 sec, and final extension at 72 °C for 10 min. The quality of PCR was tested on 1% agarose gel electrophoresis stained with ethidium bromide. PCR products were purified and sequenced with primers mentioned above by Sangon Biotech (Shanghai), Co., Ltd., China.

*2.3. Sequence Alignment and Phylogenetic Analysis*

The sequence chromatograms were checked by using BioEdit v.7.0.9 [33], and the forward and reverse sequences were assembled by SeqMan Pro in DNASTAR Lasergene v7.1 (DNASTAR, Inc. Madison, WI, USA). The reference sequences were retrieved from relevant publications (Table 1) and downloaded from GenBank. Sequences were aligned using MAFFT v.7 [34] and manually adjusted by BioEdit v.7.0.9 [33]. The single gene sequence datasets (SSU, ITS, LSU and TEF1-α) were concatenated with Mesquite v.3.70 [35] and the combined alignment was transformed into appropriate file formats for different analyzing programs by using a web server ALTER [36].

The analyses of maximum parsimony (MP), maximum likelihood (ML) and Bayesian inference (BI) were carried out as detailed in Dissanayake et al. [37]. The programs used in this study were PAUP v.4.0b 10 [38], raxmlGUI v. 1.3 [39] and MrBayes v3.1.2 [40,41]. Maximum likelihood analysis was performed by raxmlGUI v.1.3 [39] with GTR+I+G as the evolution model. In BI analyses, MrModeltest v. 2.3 [42] was used to select the best-fit model of nucleotide substitution for each partition. GTR+I+G is the best-fit model of ITS, LSU, TEF1-α, and HKY+I+G is the best model of SSU. Four simultaneous Markov chains were run for 1,000,000 generations and partition analysis with 100 sample frequencies, which produced 10,000 trees. The first 1000 trees were discarded as burn-in phase and the remaining 9000 trees were used for calculating posterior probabilities.

Phylograms were visualized by FigTree v.1.4.4 program [43] and edited with Adobe Illustrator CS5 v. 15.0.0 (Adobe®, San Jose, CA, USA). New sequences in the present study were deposited in GenBank (Table 1), and the final alignments were deposited in TreeBASE (www.treebase.org, accessed on 4 November 2021) with a submission number: 28957.

**Table 1.** Taxa used in this study and their GenBank accession numbers. Newly generated sequences are indicated with * and the ex-type strains are in bold.

| Taxa | Strain/Voucher | LSU | ITS | SSU | TEF1-α | References |
|---|---|---|---|---|---|---|
| *Bambusicola bambusae* | MFLUCC 11-0614 | JX442035 | NR121546 | JX442039 | KP761722 | [44] |
| *Bambusicola irregulispora* | MFLUCC 11-0437 | JX442036 | NR121547 | JX442040 | KP761723 | [25] |
| ***Bimuria novae-zelandiae*** | **CBS 107.79** | **AY016356** | – | **AY016338** | **DQ471087** | [14] |
| ***Darksidea alpha*** | **CBS 135650** | **KP184019** | **KP183998** | **KP184049** | **KP184166** | [9] |
| ***Darksidea beta*** | **CBS 135637** | **KP184023** | **KP183978** | **KP184074** | **KP184189** | [9] |
| ***Darksidea delta*** | **CBS 135638** | **KP184024** | **KP183981** | **KP184069** | **KP184184** | [9] |
| ***Darksidea epsilon*** | **CBS 135658** | **KP184029** | **KP183983** | **KP184070** | **KP184186** | [9] |
| ***Darksidea gamma*** | **CBS 135634** | **KP184028** | **KP183985** | **KP184073** | **KP184188** | [9] |
| ***Darksidea zeta*** | **CBS 135640** | **KP184013** | **KP183979** | **KP184071** | **KP184191** | [9] |
| ***Didymosphaeria rubi-ulmifolii*** | **MFLUCC 14-0023** | **KJ436586** | – | **KJ436588** | – | [14] |
| *Crassoascoma potentillae* | UESTCC 21.0010 * | OK161254 | OK161237 | OK161233 | OK181165 | This study |
| *Crassoascoma potentillae* | UESTCC 21.0011 * | OK161255 | OK161238 | OK161234 | OK181166 | This study |

**Table 1.** *Cont.*

| Taxa | Strain/Voucher | LSU | ITS | SSU | TEF1-α | References |
|---|---|---|---|---|---|---|
| *Crassoascoma potentillae* | UESTCC 21.0012 * | OK161256 | OK161239 | OK161235 | OK181167 | This study |
| *Crassoascoma potentillae* | **CGMCC 3.20483 *** | **OK161257** | **OK161240** | **OK161236** | **OK181168** | This study |
| **Halobyssothecium carbonneanum** | **CBS 144076** | **MH069699** | **MH062991** | – | – | [45] |
| **Halobyssothecium estuariae** | **MFLUCC 19-0386** | **MN598871** | **MN598890** | **MN598868** | **MN597050** | [46] |
| *Halobyssothecium kunmingense* | KUMCC 19-0101 | MN913732 | MT627715 | MT864313 | – | [47] |
| **Halobyssothecium obiones** | **MFLUCC 15-0381** | **MH376744** | **MH377060** | **MH376745** | **MH376746** | [10] |
| **Halobyssothecium unicellulare** | **MD 6004** | **KX505376** | – | **KX505374** | – | [48] |
| **Halobyssothecium voraginesporum** | **MD1342** | **KX499520** | – | **KX499519** | – | [48] |
| *Helminthosporium velutinum* | MAFF 243854 | AB807530 | LC014556 | AB797240 | AB808505 | [14] |
| **Katumotoa bambusicola** | **KT1517a** | **AB524595** | **LC014560** | **AB524454** | **AB539108** | [8] |
| **Keissleriella breviasca** | **KT649** | **AB807588** | **AB811455** | **AB797298** | **AB808567** | [14] |
| **Keissleriella caraganae** | **KUMCC 18-0164** | **MK359439** | **MK359434** | **MK359444** | **MK359073** | [47] |
| *Keissleriella cladophila* | CBS 104.55 | GU301822 | – | GU296155 | GU349043 | [14] |
| *Keissleriella culmifida* | KT2642 | AB807592 | LC014562 | AB797302 | AB808571 | [14] |
| *Keissleriella genistae* | CBS 113798 | GU205222 | – | GU205242 | – | [14] |
| *Keissleriella gloeospora* | KT829 | AB807589 | LC014563 | AB797299 | AB808568 | [14] |
| **Keissleriella poagena** | **CBS136767** | **KJ869170** | **KJ869112** | – | – | [14] |
| **Keissleriella quadriseptata** | **KT2292** | **AB807593** | **AB811456** | **AB797303** | **AB808572** | [14] |
| *Keissleriella taminensis* | KT571 | AB807595 | LC014564 | AB797305 | AB808574 | [14] |
| **Keissleriella trichophoricola** | **CBS 136770** | **KJ869171** | **KJ869113** | – | – | [14] |
| **Keissleriella yonaguniensis** | **KT 2604** | **AB807594** | **AB811457** | **AB797304** | **AB808573** | [14] |
| **Lentithecium clioninum** | **KT1149A** | **AB807540** | **LC014566** | **AB797250** | **AB808515** | [14] |
| *Lentithecium fluviatile* | CBS 122367 | GU301825 | – | GU296158 | GU349074 | [14] |
| **Lentithecium pseudoclioninum** | **KT1113** | **AB807545** | **AB809633** | **AB797255** | **AB808521** | [14] |
| **Magnicamarosporium iriomotense** | **KT2822** | **AB807509** | **AB809640** | – | **AB808485** | [14] |
| *Massarina eburnea* | CBS 473.64 | GU301840 | – | GU296170 | GU349040 | [14] |
| *Montagnula opulenta* | CBS 168.34 | DQ678086 | – | AF164370 | – | [14] |
| **Murilentithecium clematidis** | **MFLUCC 14-0562** | **KM408759** | **KM408757** | **KM408761** | **KM454445** | [13] |
| **Murilentithecium rosae** | **MFLUCC 15-0044** | **MG829030** | **MG828920** | **MG829137** | – | [49] |
| **Neobambusicola strelitziae** | **CBS 138869** | **KP004495** | – | – | – | [14] |
| **Neoophiosphaerella sasicola** | **KT1706** | **AB524599** | **LC014577** | **AB524458** | **AB539111** | [14] |
| **Phragmocamarosporium platani** | **MFLUCC 14-1191** | **KP842916** | – | **KP842919** | – | [15] |
| **Phragmocamarosporium rosae** | **MFLUCC 17-0797** | **MG829051** | – | **MG829156** | **MG829225** | [49] |
| **Pleurophoma ossicola** | **CPC 24979** | **KR476769** | **KR476736** | – | – | [16] |

**Table 1.** *Cont.*

| Taxa | Strain/Voucher | LSU | ITS | SSU | TEF1-α | References |
|---|---|---|---|---|---|---|
| *Pleurophoma pleurospora* | **CBS130329** | **JF740327** | – | – | – | [14] |
| *Poaceascoma aquaticum* | **MFLUCC 14-0048** | **KT324690** | – | **KT324691** | – | [50] |
| *Poaceascoma filiforme* | **CBS 146689** | **MT373345** | **MT373362** | – | – | [51] |
| *Poaceascoma halophila* | **MFLUCC 15-0949** | **MF615399** | – | **MF615400** | – | [52] |
| *Poaceascoma helicoides* | **MFLUCC 11-0136** | **KP998462** | **KP998459** | **KP998463** | **KP998461** | [17] |
| *Poaceascoma taiwanense* | **MFLU 18-0083** | **MG831567** | **MG831569** | **MG831568** | – | [23] |
| *Setoseptoria arundelensis* | **MFLUCC 17-0759** | **MG829073** | **MG828962** | **MG829173** | – | [49] |
| *Setoseptoria arundinacea* | KT600 | AB807575 | LC014595 | AB797285 | AB808551 | [14] |
| *Setoseptoria englandensis* | **MFLUCC 17-0778** | **MG829074** | **MG828963** | **MG829174** | – | [49] |
| *Setoseptoria macropycnidia* | CBS114202 | GU301873 | – | GU296198 | GU349026 | [14] |
| *Setoseptoria magniarundinacea* | **KT1174** | **AB807576** | **LC014596** | **AB797286** | **AB808552** | [14] |
| *Setoseptoria phragmitis* | **CBS 114802** | **KF251752** | **KF251249** | – | – | [19] |
| *Setoseptoria scirpi* | **MFUCC 14-0811** | **KY770982** | **MF939637** | **KY770980** | **KY770981** | [52] |
| *Spegazzinia deightonii* | MAFF 243876 | AB807581 | – | AB797291 | AB808557 | [14] |
| *Sulcatispora acerina* | **KT 2982** | **LC014610** | **LC014597** | **LC014605** | **LC014615** | [14] |
| *Sulcatispora berchemiae* | **KT 1607** | **AB807534** | **AB809635** | **AB797244** | **AB808509** | [14] |
| *Tingoldiago clavata* | **MFLUCC 19-0496** | **MN857178** | **MN857182** | **MN857186** | – | [25] |
| *Tingoldiago graminicola* | **KH68** | **AB521743** | **LC014598** | **AB521726** | **AB808561** | [14] |
| *Tingoldiago hydei* | **MFLUCC 19-0499** | **MN857177** | **MN857181** | – | – | [25] |
| *Towyspora aestuari* | **MFLUCC 15-1274** | **KU248852** | **KU248851** | **KU248853** | – | [21] |
| *Wettsteinina lacustris* | CBS 618.86 | – | AF250831 | DQ678023 | DQ677919 | [8] |

### 3. Results

*3.1. Phylogenetic Analyses*

The final dataset comprised of 67 taxa selected from *Bambusicolaceae, Didymosphaeriaceae, Lentitheciaceae, Massarinaceae* and *Sulcatisporaceae* in the suborder *Massarineae* (*Pleosporales, Dothideomycetes*), with *Helminthosporium velutinum* (MAFF 243854) and *Massarina eburnean* (CBS 473.64) as outgroup taxa. The combined dataset consisted of 4361 characters after alignment including gaps (SSU: 1423 bp; ITS: 731 bp; LSU: 1283 bp; TEF-1α: 924 bp). ML, MP and BI analyses were conducted and resulted in generally congruent topologies. The best scoring ML tree (Figure 1) with a final optimization likelihood value of $-22,480.205214$. The aligned matrix had 1375 distinct alignment patterns, and 34.57% completely undetermined characters and gaps. Estimated base frequencies were as follows: A = 0.240681, C = 0.247239, G = 0.272357, T = 0.239543; substitution rates AC = 1.235890, AG = 2.828734, AT = 1.375693, CG = 1.247991, CT = 7.529110, GT= 1.000000; gamma distribution shape parameter α = 0.468984. Maximum parsimony analyses indicated 3258 constant characters and included 317 variable characters of parsimony-uninformative and 786 characters were parsimony informative. A heuristic search yield one equally most parsimonious trees (TL = 3250, CI = 0.487, RI = 0.670, RC = 0.326, HI = 0.513).

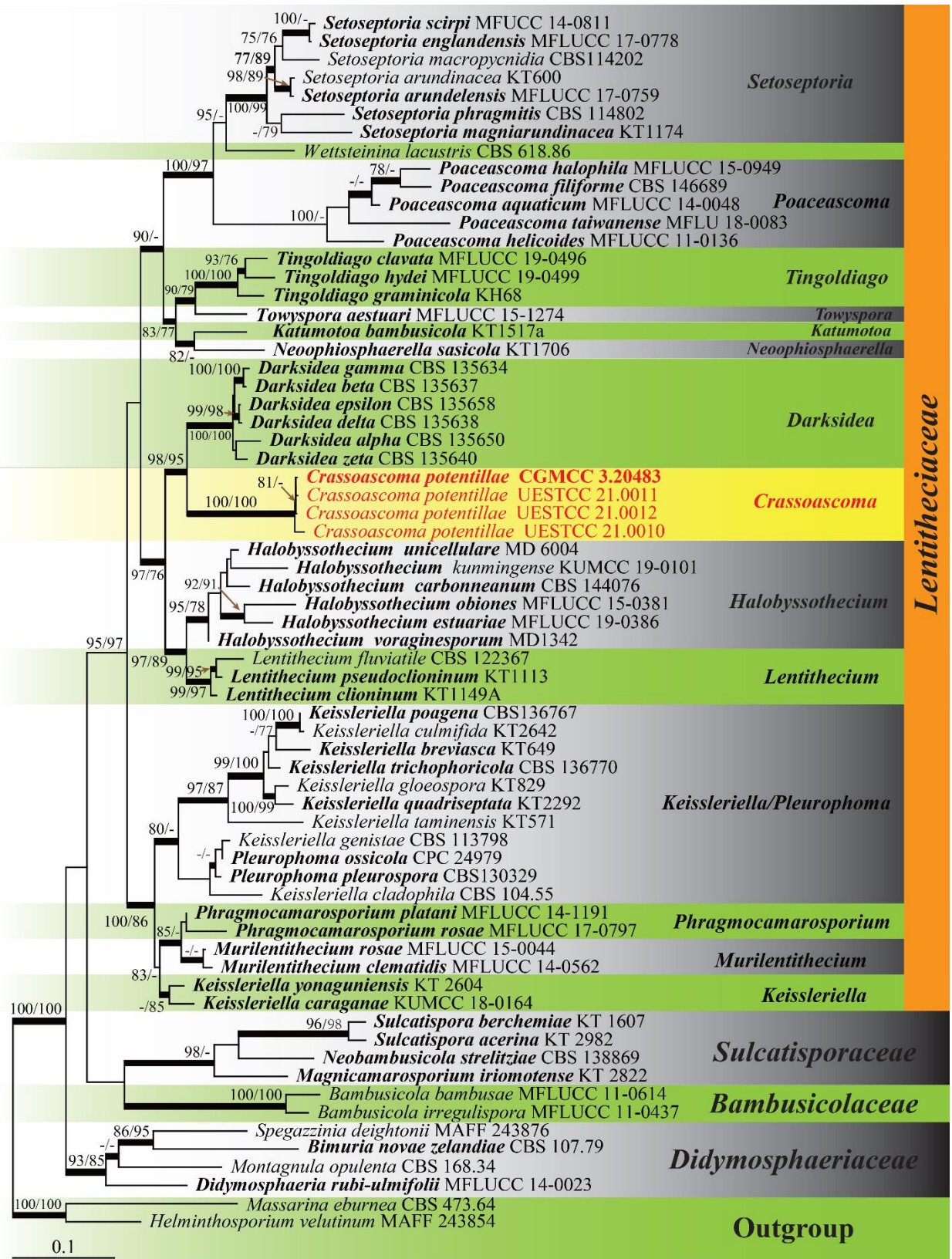

**Figure 1.** RAxML analysis of *Lentitheciaceae* and representatives in *Massarineae* based on combined SSU ITS, LSU, and TEF1-α sequence data. Bootstrap support values for maximum likelihood (ML) and maximum parsimony (MP) higher than 75% were placed above the branches (ML/MP). Bayesian posterior probabilities (BYPP) greater than 0.95 were shown as bold branches. Ex-type strains were in bold and new strains generated in this study were indicated in red.

Fourteen genera are accepted in the family *Lentitheciaceae* and all have available sequence data [1]. However, the genus *Pseudomurilentithecium* [18] was excluded from our final dataset because it was unstable in the single gene phylogenetic analyses. *Pseudomurilentithecium* fell outside *Lentitheciaceae* in the LSU and TEF1-α gene trees (data not shown), but it formed an obvious long branch in this family in the multi-gene phylogeny result. The four newly generated isolates formed a well-supported monophyletic clade in *Lentitheciaceae* (Figure 1) and can be recognized as a new genus *Crassoascoma*. Moreover, the molecular data of these four isolates are identical, which supports the identification of a new species namely *C. potentillae*.

*3.2. Taxonomy*

*Crassoascoma* Jian K. Liu, gen. nov
*MycoBank*: MB 841098
*Etymology*: Referred to superficial thick-walled ascoma.
*Saprobic* on the living and decayed branches of *Potentilla fruticosa* L. (*Rosaceae*). Sexual morph: *Ascomata* superficial, solitary to gregarious, subglobose to globose, dark brown to black, ostiolate, with setae around the papilla. *Peridium* with multi-layers, comprising hyaline to brown cells of textura angularis. *Hamathecium* trabeculate pseudoparaphyses, filamentous and anastomosing. *Asci* 8-spored, clavate to subcylindrical, short pedicel. *Ascospores* overlapping biseriate, usually uniseriate in the lower half, fusiform, straight or slightly curved, hyaline to pale brown, 1-septate constricted at the septum and midpoint of each cell, surrounded by an entire mucilaginous sheath. Asexual morph: Undetermined.
*Type species*: *Crassoascoma potentillae* Z.P. Liu, S.N. Zhang and Jian K. Liu
*Notes*: The phylogenetic analyses showed that four isolates of *Crassoascoma* formed a monophyletic clade in *Lentitheciaceae* and is sister to *Darksidea* with high statistical support (98/95/1.00). *Darksidea* accommodates endophytic fungi which are characterized by 4–6-spored, clavate to ellipsoid asci and ellipsoid, aseptate ascospores [9]. *Crassoascoma* however, is distinct from *Darksidea* in having superficial, papillate, thick-walled ascomata with setae around the ostiole, clavate to subcylindrical asci, and fusiform, 1-septate ascospores surrounded by a mucilaginous sheath. Multi-gene phylogeny results also indicated a close relationship of *Crassoascoma* to *Lentithecium* and *Halobyssothecium*. The sexual morphs of these three genera are somewhat similar, and all have eight-spored, clavate to subcylindrical asci, fusiform ascospores with septa that constricted at median septum. However, *Lentithecium* contains saprobic species from aquatic habitats, usually have semi-immersed, thin-walled ascomata, and fusiform, hyaline, or yellowish-brown ascospores, with or without a sheath [8,53,54]. The superficial, black, thick-walled ascomata of *Crassoascoma* resemble species in *Halobyssothecium* [10,55], but they can be distinguished by the ascospores (hyaline to pale brown vs. versicolored; with sheath vs. lacking sheaths or appendages) and habitat (terrestrial semiarid region vs. marine).
*MycoBank*: MB 841099
*Etymology*: Name "*potentillae*" referred to the hosts *Potentilla fruticosa* L. on which the fungus was collected.
*Holotype*: HKAS 115966
*Crassoascoma potentillae* Z.P. Liu, S.N. Zhang and Jian K. Liu, sp. nov. Figure 2
*Saprobic* on the epidermis of decayed wood in semiarid lands. Sexual morph: *Ascomata* 270–340 μm high, 270–320 μm diam. superficial, scattered, subglobose to globose, base flattened, dark brown to black, uni-loculate, rough walled, papillate. *Ostiole* 90–110 μm wide, 100–120 μm high, central, dark brown, with brown setae on the papilla. *Peridium* 28–37 μm wide, relatively thick, multi-layer, comprising pale brown to brown cells of textura angularis. *Hamathecium* 1.7–2.3 μm wide, trabeculate pseuoparaphyses, remotely septate, filamentous and anastomosing. *Asci* 139–155 × 18–20 μm (x̄ = 147 × 17 μm, n = 20), 8-spored, bitunicate, fissitunicate, clavate to subcylindrical, shortly pedicellate, apically rounded with an ocular chamber. *Ascospores* 31–35 × 7–8 μm (x̄ = 33 × 6.5 μm, n = 50), overlapping biseriate, usually uniseriate in the lower half, fusiform with subobtuse to acute ends, straight or

slightly curved, hyaline, rarely pale brown, one-septate, the upper cell slightly bigger than the lower cell, constricted at the septum and midpoint of each cell, smooth-walled, surrounded by an entire mucilaginous sheath. Asexual morph: undetermined.

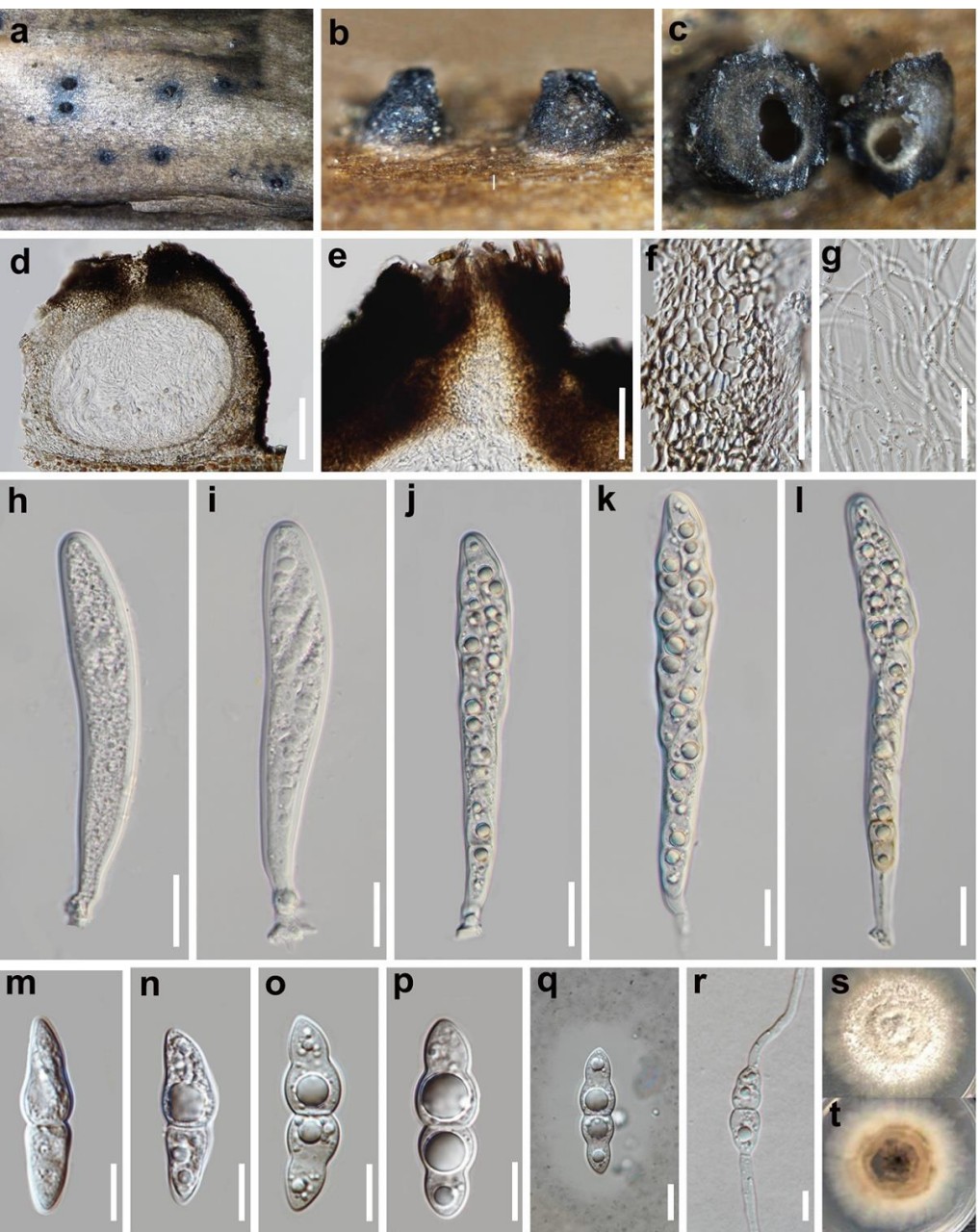

**Figure 2.** *Crassoascoma potentillae* (HKAS 115966, holotype) (**a**–**c**) Ascomata on host surface. (**d**) Vertical section of ascomata. (**e**) Ostiole. (**f**) Peridium. (**g**) Hamathecium. (**h**–**l**) Asci, (**m**–**p**) Ascospores. (**q**) Ascospore stained with India ink showing the mucilaginous sheath. (**r**) Germinating ascospore. (**s**,**t**) Colony on PDA (4 weeks). Scale bars: d = 100 μm, e = 50μm, (**f**–**l**), r = 20 μm, (**m**–**q**) = 10 μm.

*Culture characteristics:* Ascospores germinated on PDA within 24 h. Colonies on PDA reaching 24 mm diam. incubated at 25 °C after 4 weeks, circular, medium dense, cottony, flat to raised, entire margins, greyish-white, thinner and velvety at the margin, pale brown to brown from the reverse.

*Material examined*: CHINA. Qinghai province, Huangnan autonomic prefecture, Zeku city, Maixiang district S203 road, 35°16′27″ N, 101°54′48″ E, 3315 m, on the branch of *Potentilla fruticosa* L. (*Rosaceae*), 21 July 2020, Jian-Kui Liu, HUEST 21.0015 (HKAS 115966,

holotype; ex-type living culture CGMCC 3.20483 = UESTCC 21.0013); *ibid.*, Gansu Province, Zhangye city, Minle county, 38°23′2″ N, 100°22′30″ E, 2830 m, on the branch of *Potentilla fruticosa* L., 2 August 2020, Jian-Kui Liu, HUEST 21.0013 (HKAS 115967, paratype); living culture UESTCC 21.0011; *ibid.*, Gansu Province, Zhangye City, Minle County, 38°23′2″ N, 100°22′30″ E, 2830 m, on the branch of *Potentilla fruticosa* L., 2 August 2020, Jian-Kui Liu, HUEST 21.0014; living culture UESTCC 21.0012; *ibid.*, Gansu Province, Jiuquan City, Sunan County, Kangle Township, 38°47′50″ N, 99°40′53″ E, 2456 m, on the branch of *Potentilla fruticosa* L., 2 August 2020, Jian-Kui Liu, HUEST 21.0012, living culture UESTCC 21.0010.

*Notes*: Four of our *Crassoascoma* isolates clustered together and almost identical in their ITS, TEF1-α sequence data, which indicated they can be recognized as a single species. The distinguishing characteristics of *Crassoascoma potentillae* are the superficial, globose, black, thick-walled ascomata with setae on the papilla, and fusiform, one-septate, hyaline (rarely pale brown) ascospores with subobtuse ends, constricted at the septum and midpoint of each cell and surrounded by a mucilaginous sheath.

## 4. Discussion

This study contributed to the microfungal community in semiarid lands on the Qinghai-Tibet Plateau, of which a monotypic genus *Crassoascoma* belongs to *Lentitheciaceae* (*Dothideomycetes*) was isolated, identified and well-described. Although the species of *Lentitheciaeae* are morphologically diverse, this new genus is unique in terms of the superficial, thick-walled ascomata with setae around the papilla. Along with the ascospores morphology, *Crassoascoma* can be clearly distinguished from other genera in *Lentitheciaceae*. The discovery of the interesting new genus in the special environmental habitats (e.g., high-elevation, dry, low temperate) also reflects that there may be many fungal novelties in the Qinghai-Tibet Plateau to be explored, discovered and described.

**Author Contributions:** Conceptualization, J.-K.L., Z.-P.L. and S.-N.Z.; methodology, S.-N.Z.; formal analysis, Z.-P.L.; resources, J.-K.L.; data curation, Z.-P.L. and R.C.; writing—original draft preparation, Z.-P.L. and S.-N.Z.; writing—review and editing, Z.-P.L.; S.-N.Z. and J.-K.L.; supervision, J.-K.L.; project administration, Q.Z. and J.-K.L.; funding acquisition, Q.Z. All authors have read and agreed to the published version of the manuscript.

**Funding:** This study was funded by the second Tibetan Plateau Scientific Expedition and Research (STEP) Program, Grant Number 2019QZKK0503.

**Institutional Review Board Statement:** Not applicable.

**Data Availability Statement:** The data are available upon request from the corresponding author. The data is not publicly available due to it its usage in the ongoing study.

**Acknowledgments:** Z.-P.L. thanks Na Wu for her assistance in fungal culture isolation and illustration.

**Conflicts of Interest:** The authors declare no conflict of interest.

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
