# Peer review of "Crassoascoma gen. nov. (Lentitheciaceae, Pleosporales): Unrevealing Microfungi from the Qinghai-Tibet Plateau in China"

_diversity, doi:10.3390/d14010015_

Round 1
Reviewer 1 Report
Aside from the Abstract, in which the authors fail to highlight the importance of the study, this manuscripts represents a well performed and described study on the isolation and characterization of a new genus of microfungi.
I think there is also a minor typo in the title: the the authors mean "Revealing" rather than "Unrevealing"? I would suggest making this change.
Author Response
Reviewer #1: Aside from the abstract, in which the authors fail to highlight the importance of the study, this manuscript represents a well performed and described study on the isolation and characterization of a new genus of microfungi.
Our response: Thanks very much for your kind comments. We have improved the manuscript and emphasized the importance of the study in the discussion part. Please kindly see the revised version.
Reviewer #1: I think there is also a minor typo in the title: the authors mean “Revealing” rather than “Unrevealing”? I would like to suggest making this change.
Our response: Thanks very much for your suggestion. As the two words represent a similar meaning in the title and we prefer to keep “unrevealing” as it is more suitable to indicate the status of fungal studies in Qinghai-Tibet Plateau (its unique geography and habitats).
Reviewer 2 Report
Dear authors,
this work is interesting and useful for the scientific community working on microfungi. This work is well presented. However, changes must be incorporated before publication.
Please organize your section 3.2 taxonomy better,
Line 165-167, 194-198: make sentences
Line 177: make a sentence
Line 178, 213, 233: make a sentence
Line 222: material examined: make a sentence .......
Make a structured and organized paragraph
There is no conclusion and no discussion, please add them.
Line 239,244,247,248: remember to remove the yellow highlighting thank you
Author Response
Reviewer #2: Please organize your section 3.2 taxonomy better, Line 165-167, 194-198: make sentences.
Our response: Thanks very much for your comments. As the general format requirement, we follow the guideline of the taxonomy descriptions and illustration, therefore we keep them as the current format.
Reviewer #2: Line 177: make a sentence.
Our response: same as our answer above (our response to Line 165-167, 194-198).
Reviewer #2: Line 178, 213, 233: make a sentence.
Our response: same as our answer above (our response to Line 165-167, 194-198). Line 178 and line 233 start with “notes”, which is a paragraph that discussed the recognition of the new genus and the new species, respectively. Line 213 starts with “Culture characteristics”, which is a paragraph that described the fungal colony.
Reviewer #2: Line 222: make a sentence…make a structured and organized paragraph.
Our response: same as our answer above (our response to Line 165-167, 194-198). This paragraph listed the specimen information.
Reviewer #2: There is no conclusion and no discussion, please add them.
Our response: Thanks for your suggestions. We have considered this and added the discussion, please kindly see the revised file.
Reviewer #2: Line 239, 244, 247, 248: remember to remove the yellow highlighting.
Our response: yes, thank you for pointing out this. we have removed the yellow highlighting.
Reviewer 3 Report
Dear authors,
please check comments inside attached document.

Author Response
Reviewer #3: Line 14, “we are participating in an investigation….”. I would rewrite first 2 phrases to direct the message clearly; example: “Microfungal communities associated with…QTP were studied and four collections assoc with P. fructicosa were obtained…China”
Our response: Thanks very much for your comments and suggestion. We have improved it as “Microfungi associated with woody plants on the Qinghai-Tibet Plateau (QTP) were investigated, and four collections associated with Potentilla fruticose were obtained from Gansu and Qinghai provinces in China”. Please kindly see the revised file.
Reviewer #3: Line 22, “sister to Darksidea”, “sister clade”
Our response: Thanks for your comment. We have corrected it as “sister clade to Darksidea”. Please kindly see the revised version.
Reviewer #3: Line 77, “slides within water mounts”, “slides mounted in water”
Our response: Thanks for your comment. We have corrected it as “slides mounted in water”. Please kindly see the revised version.
Reviewer #3: Line 104, “for 30sec,”
Our response: We have corrected it as “for 30 sec,”, please kindly see the revised version.
Reviewer #3: Line 239, 244, 247, 248, delete colours.
Our response: We have deleted the colours, please kindly see the revised version.